# Heme Oxygenase-1 and Blood Bilirubin Are Gradually Activated by Oral D-Glyceric Acid

**DOI:** 10.3390/antiox11122319

**Published:** 2022-11-23

**Authors:** O. Petteri Hirvonen, Maarit Lehti, Heikki Kyröläinen, Heikki Kainulainen

**Affiliations:** Faculty of Sport and Health Sciences, Neuromuscular Research Center, University of Jyväskylä, 40014 Jyväskylä, Finland

**Keywords:** HO-1, bilirubin, HIF-1α, subclinical inflammation, ROS

## Abstract

It has been shown that small doses of oral D-glyceric acid (DGA) activate mitochondrial metabolism and reduce inflammation among 50–60-year-old healthy volunteers. The present results with the same small doses reveal that after a 4-day DGA regimen, a dose of DGA activated the HO-1 pathway acutely, while enhanced inflammatory status after the 4-day DGA regimen seemed to be able to downregulate the HO-1 pathway in non-acute measurement. Blood bilirubin was strongly upregulated towards the end of the altogether 21-day study period with positive associations towards improved inflammation and reduced blood triglycerides. After the 4-day DGA regimen, hepatic inflow of blood bilirubin with albumin as the carrier was clearly upregulated in the lower-aerobic-capacity persons. At the same time also, blood triglycerides were down, pointing possibly to the activation of liver fatty acid oxidation. The combination of activated aerobic energy metabolism with transient HO-1 pathway activation and the upregulation of blood bilirubin may reduce the risks of chronic diseases, especially in aging. Furthermore, there exist certain diseases with unsatisfactorily-met medical needs, such as fatty and cholestatic liver diseases, and Parkinson’s disease, that can be possibly ameliorated with the whole-body mechanism of the action of the DGA regimen.

## 1. Introduction

Bilirubin is a product of heme catabolism. The cytosolic heme oxygenase (HO) reaction produces biliverdin, iron (Fe), and carbon monoxide (CO) [1,2]. Biliverdin is further reduced into (unconjugated) bilirubin via the biliverdin reductase (BVR) enzyme [2]. In the BVR reaction, one cytosolic NADH is oxidized into NAD^+^. The majority of blood bilirubin is made from the breakdown of hemoglobin from senescent red blood cells, but a significant part of bilirubin also originates from the turnover of various heme-containing proteins found in other tissues, primarily in the liver and muscles. Heme oxygenase is the rate-limiting factor in bilirubin production [3].

Water insoluble unconjugated bilirubin is excreted from cells and carried by albumin in plasma for hepatic conjugation and further excretion via bile ducts into the intestines [4]. An increase in bilirubin concentration, both in plasma and in tissues, has been shown to possess significant antioxidant [5] and anti-inflammatory effects, as well as therapeutic effects in neurodegenerative diseases, such as Parkinson’s disease [6,7]. Additionally in recent studies, bilirubin has been shown to activate fatty acid metabolism via the peroxisome proliferator-activated receptor alpha (PPARα) [8].

Inducible heme oxygenase (HO-1) activity has also been found relevant for antioxidant and anti-inflammatory protection independently of bilirubin. Furthermore, it is beneficial in therapies of many chronic diseases, such as NAFLD [9]. HO-1 is activated by stress factors e.g., by increased reactive oxygen species (ROS) generation via the NF-E2–related factor2 (Nrf2 transcription factor). The Nrf2 transcription factor upregulates the mRNA, protein, and enzymatic activity of HO-1 [10]. Nrf2/HO-1 is induced, e.g., by oxidative stress from mitochondrial oxidative phosphorylation (OXPHOS). Nrf2 related HO-1 activation is additionally linked to the upregulation of mitochondrial biogenesis [10]. For simplicity, we later refer to combined HO-1 and BVR reactions that generate bilirubin and Fe as the HO-1 pathway. 

Too much Nrf2 or HO-1 activation may also cause deleterious effects [1]. In fact, all the products of the HO-1 pathway are toxic at higher concentrations, but even at physiological levels CO may possess important protective effects in certain pathologic situations [11,12]. Endogenous CO is eventually excreted from the body via the lungs [13]. The liver stores and regulates excess endogenous blood Fe [14].

HO-1 is upstream of bilirubin while the hypoxia inducible factor 1 alpha (HIF-1α) expression is induced by bilirubin in the physiological oxygen content [15], i.e., HIF-1α may be downstream of bilirubin when bilirubin concentration is increased in normoxia. Furthermore, HO-1 has been recognized as a downstream gene of HIF-1α pathway in many tissues during hypoxia [16,17]. Finally, HO-1 expression has been shown to be transcriptionally regulated by PPARα [18]. Due to multiple direct and indirect connections and feedback mechanisms, it is difficult or sometimes even impossible to separate independent therapeutic effects of HO-1 activation from the effects of bilirubin, and vice versa [19]. This is especially valid in studies related to whole physiological systems.

Physical exercise activates ATP generation by cytosolic glycolysis and mitochondrial OXPHOS, and also ROS generation [20]. Induced ROS generation activates a counterbalancing antioxidant HO-1 pathway, and as a follow-up the generation of bilirubin and Fe are upregulated. It has been found that conducting strenuous exercises elevates the plasma bilirubin level [21]. Furthermore, some studies indicate that plasma bilirubin is elevated in highly exercising subjects due to the reduced rate of hepatic bilirubin conjugation and subsequent reduced excretion of bilirubin from the body [22,23].

Effective counterbalancing of elevated oxidative stress and inflammation makes HO-1 pathway studies somewhat challenging because HO-1 activation aims at suppressing itself via neutralizing ROS that originally caused the activation. Additionally, all details of bilirubin in the entire body metabolism are not fully understood [7]. Nevertheless, it is wildly accepted that a gradual basal increase in bilirubin concentration within the physiological range and mild or transient HO-1 activation forms therapeutic effects throughout the body [24,25].

D-glyceric acid (DGA) is a trace metabolite present in vertebrates [26]. An oral DGA regimen has been shown to possess similar signaling effects to physical exercise and may promote health benefits related to mitochondrial activation, at least in older persons [27].

In this study we aim to find out whether the HO-1 pathway is activated and the blood bilirubin level is elevated after an acute oral DGA dose. The same research questions are studied in the non-acute 4-day and 4 + 14-day DGA administration of doses in healthy 50–60-year-old humans. Additionally, shorter 4-day whole-body effects of the DGA regimen on the HO-1 pathway are analyzed separately in high-capacity and lower-capacity persons. Because possible ROS scavenging (antioxidant) effects can be analyzed only in cell cultures, we report the impact of DGA administration on ROS generation from a human primary hepatocyte study and from a rat primary astrocyte study in the Appendix A.

## 2. Materials and Methods

### 2.1. Participants

Altogether 27 healthy 50–60-year-old Caucasian participants were carefully selected for the study group. This age group was chosen because systemic inflammation markers are on average somewhat elevated even in healthy persons at that age [28]. All the participants were informed of the experimental design, and the benefits and possible risks that could be associated with the study prior to signing an informed consent to voluntarily participate in the study. Detailed selection criteria and other characteristics beyond the information presented in Table 1B can be found in Hirvonen et al., (2021) [27]. All studies were conducted in line with the statement of the Ethical Committee of the Central Finland Health Care District (Dnro 1U/2019, KSSHP). Trial registration number (14 January 2021), ClinicalTrials.gov Identifier: NCT04713319.

### 2.2. Study Setup and Monitoring

There were altogether three measurement days (Day0, Day4 and Day21) in the study (Table 1A). First fasting and resting blood samples at Day0, Day4 and Day21 were taken 12 h after the last DGA or placebo dose (non-acute). Blood samples for each participant were always taken at the same time in the morning (±2 min). On Day4, an acute blood sample was additionally drawn at the study site 45 min after the morning sample. In between, a dose of DGA or placebo was taken immediately after the morning blood sample (Table 1A). Furthermore, on Day0 after the morning blood sample, an indirect VO_2_max test with a bicycle ergometer was performed for all participants to assess aerobic capacity.

The test setup was double-blinded. Measurements were always performed on the same weekday (Friday or Saturday) for each participant. To avoid any bias from the strenuous VO_2_max test at Day0, we added two full recovery days after it before initiating the oral DGA regimen. For simplicity, we call the second measurement day “Day4” because it was taken after the 4-days of the DGA regimen. The different phases of the study are illustrated in Table 1A.

Normal lifestyle and stable behavior were encouraged through the use of personal diaries and reminder emails. Morning interviews were carried out individually when participants arrived at the study site at a minimum of 30 min before the first blood sample. Moreover, participant’s health status and the timing of the last dose were always checked. All the participants, who came to the Day0 measurement, completed the whole study.

### 2.3. Characteristics of the Study Group and Analysed Subgroups

The study group consisted of healthy 50–60-year-old males and females. Its characteristics are presented on the first column of Table 1B (later also called the “whole group”). In the Day4 analyses, the whole group was further divided into the high aerobic capacity (HC) and lower aerobic capacity (LC) subgroups based on seven scale classification [29]. The HC subgroup consisted of the two best aerobic capacity classes with “Excellent” and “Very Good” aerobic capacity and the LC subgroup consisted of participants from five lower classification groups. From our study group, ten participants (6 females and 4 males) ended in the HC subgroup and 17 participants (10 females and 7 males) in the LC subgroup (Table 1B).

The placebo group was chosen randomly among females and males separately beforehand. In practice, the number of placebo-treated participants was zero until Day4 morning measurement, which enabled HC–LC subgroup division with a sufficient number of observations for statistical comparison. The priming of the placebo group with the 4-day DGA regimen may have had an impact on the 45-min results on Day4, and this was considered when analyzing them.

### 2.4. Test Substances and Doses, Preparations, and Administration

D-glyceric acid calcium salt and the placebo (E509/calcium chloride) were dissolved into 1.5 L bottles of water beforehand for each participant. The calculated dose of DGA or placebo was to be drunk in the morning and in the evening. In the placebo group there were equal molar amounts of calcium with water.

Selected doses, regimens, and measurement timings in the present study were based on earlier in vitro and in vivo pre-tests related to the regulatory acceptance processes of DGA. The dose of DGA used was 2 times 3.33 mg per body mass (kg) per day. The acute dose of DGA received on the morning of Day4 was the same as the previous 8 doses, and the placebo group received an equimolar amount of calcium within calcium chloride, also dissolved into water. More detailed information can be found in the study of Hirvonen et al. [27].

### 2.5. Blood Samples and Biomarker Measurements

Blood samples were drawn from the antecubital vein of each participant always at the same time in the morning. The samples were immediately cooled and centrifuged in heparin plasma tubes and stored into 2 mL portions at −80 ℃. Plasma total bilirubin and Fe were measured with clinically accredited standard methods (Synlab, Helsinki, Finland). Total bilirubin represents plasma unconjugated bilirubin in our study model, and in the following, the term plasma bilirubin represents plasma total bilirubin. Plasma glycoprotein acetyls (GlycA) [30], triglycerides (TGs), free fatty acids, and albumin were measured (Nightingale Health Oy, Kuopio, Finland) using nuclear magnetic resonance (NMR) technology with regulatory approval for diagnostics [31]. GlycA, TGs, free fatty acids, and albumin were measured only from non-acute samples. Plasma interleukin 6 (IL-6) was measured using the IMMULITE^®^ 2000 immunoassay system and alkaline phosphatase (ALP) with the IFCC method (kits 981832–981833, Thermo Scientific). The mRNA expressions of inducible heme oxygenase (HO-1) and hypoxia inducible factor 1 alpha (HIF-1α) were measured from white blood cells (WBCs). More detailed information can be found in Hirvonen et al. [27].

### 2.6. In Vitro Side Studies with Human Primary Hepatocytes and Rat Primary Cortical Astrocytes

In our studies with human primary hepatocytes and rat primary astrocytes, net ROS generation with DGA was compared to 0-controls, and additionally to the positive antioxidant controls [32]. On top of negative and positive controls, dose response studies were conducted to confirm consistent results.

We used two different reagents to detect oxidation caused by ROS to avoid possible pitfalls from selection of only a single detection method. In the hepatocyte studies we used a reactive oxygen species (ROS) assay kit ab113851 (Abcam). It uses the cell permeant reagent 2′,7′—dichlorofluorescein diacetate (DCFDA), a fluorogenic dye that measures hydroxyl, peroxyl and other ROS activity within the cell. In the astrocyte study we used CellROX Green staining. CellROX^®^ Oxidative Stress Reagents are fluorogenic probes designed to reliably measure ROS in live cells.

In both hepatocytes and astrocytes, mild metabolic stress was induced by new nutrition i.e., by measuring cellular ROS generation 1–2 h after the change of culture media with new nutrition and tested concentrations. The main results, including a brief methods description, are now for the first time scientifically reported in Appendix A.

### 2.7. Statistical Methods

Each person was used as her/his own control. In that way, individual “noise factors” could be eliminated and we were able to pairwise test whether the intra-individual responses were similar among the persons of the study. Furthermore, to achieve statistically unambiguous results all blinded participants were in the same comparison group during the first week. Pre-assessment on the sufficient group size was based on the results from our pilot tests and related volatilities. Among our relatively homogenous study group with fully comparable intra-individual measurement points, selected group size turned out to be clearly sufficient.

Statistical tests were conducted using IBM SPSS statistics software (ver. 26) and Microsoft Excel. Statistical tests for group averages of intra-individual changes were based on paired Student’s *t*-tests. Unpaired t-tests were used when testing the HC and LC subgroups or the DGA and placebo groups against each other. When testing intraday 45 min changes in the DGA group, possible circadian variation was corrected by deducting the mean change in the placebo group from the DGA group observations. Squared linear correlations (R^2^) from scatter diagrams were calculated between selected studied variables. Technically, R^2^ can be interpreted as the ratio of explained variation to the total variation of the explanatory variable. The significance of R^2^ was statistically tested by the F-test. When *N* < 10, we checked the normality of the underlying data visually or using the Kolmogorov–Smirnov test. A *p*-value lower than or equal to 0.05 in a one-sided t-test was considered statistically significant (marking = * or #) and a *p*-value lower than or equal to 0.01 as statistically very significant (marking = ** or ##). A non-significant *p*-value is marked by “n.s.” or > 0.05, and *p* >> 0.05 in quite clear cases. All presented tests were predetermined or derived from predetermined test settings.

## 3. Results

### 3.1. Changes in Bilirubin and Iron Concentrations during the Entire Study Period

Blood bilirubin concentration increased by 18.6% from Day0 to Day21 in the DGA administration group (*p* = 0.009) while in the placebo group, no statistically significant changes were noticed (Figure 1A). Furthermore, the Day21 changes from Day0 in the DGA group differed statistically significantly (*p* = 0.023) from the respective changes in the placebo group (Figure 1A). There was an interesting decline by 8.8% in blood Fe from Day0 to Day4 (*p* = 0.042). Blood bilirubin and Fe concentration shared a similar pattern during the study period (Figure 1A,B).

### 3.2. Biomarkers Downstream of Bilirubin and the HO-1 Pathway during the Entire Study Period

Figure 2A shows that the relative changes between GlycA and bilirubin from Day0 to Day21 correlated significantly (R^2^ = 0.353, *p* = 0.015). Respectively, the changes between triglycerides and bilirubin correlated statistically significantly (R^2^ = 0.305, *p* = 0.032) (Figure 2B) and a not-significant correlation was noticed between the changes Day4 to Day21 in IL-6 and bilirubin (Figure 2C).

Strong upregulation of blood bilirubin from Day4 to Day21 was typically accompanied by a reduction of subclinical inflammation (IL-6) at the individual level (Figure 2C). IL-6 declined by 19.0% from Day0 to Day21 in the DGA administration group (*p* = 0.002), while in the placebo group it declined only by 2.6% (Figure 2D). Furthermore, the individual changes of IL-6 in the DGA compared to the placebo group differed statistically significantly (Figure 2D) [27].

HIF-1α mRNA expression in WBCs (Figure 2E) shared a similar pattern in all observation points to blood bilirubin in the DGA group (Figure 1A). HIF-1α mRNA expression increased by 37.0% from Day0 to Day21 in the DGA administration group (*p* = 0.014) while in the placebo group there was no change (Figure 2D). Furthermore, the 49% increase (*p* = 0.004) in HIF-1α mRNA expression resembles the 24% increase in blood bilirubin from Day4 to Day21.

HO-1 mRNA expression in WBCs was very weak on Day0. Eight of the successfully analyzed twenty-five WBC samples showed a lower signal than the 0.29 detection limit and were set to zero (graph not shown). On Day4, the number of zeros increased to 11, which may suggest that the HO-1 mRNA expression was downregulated during the 4-day DGA regimen. Further in line with blood bilirubin (Figure 1A), average HO-1 mRNA expression increased in the DGA group by 34.0% on Day21 from Day0. However, poor mRNA signals destroyed the possibility of drawing statistical conclusions on the changes of HO-1 mRNA expression in WBCs.

### 3.3. Acute 45 Min Activation of the HO-1 Pathway on Day4

Blood bilirubin concentration rose by 2.3% 45 min after the acute dose in the DGA group (*p* = 0.066) but not in the placebo group (Figure 3A). Blood Fe rose by 9.1% (*p* < 0.001) in the DGA group but not in the placebo group (Figure 3B). Furthermore, there existed a statistically very significant positive correlation between the 45-min %-changes of Fe and bilirubin at the intra-individual level in the DGA group (Figure 3C).

The average relative increase in blood bilirubin (Figure 3A) was much lower than in Fe (Figure 3B) as some of the individual relative changes in blood bilirubin concentrations were negative, unlike in Fe (Figure 3C).

### 3.4. 4-Day Changes of Blood Bilirubin Are Tightly Linked to the Changes of Its Carrier in Blood

On Day0, bilirubin concentration in the HC subgroup was on average 12.8 μmol/L, while in the LC subgroup it was 9.6 μM (*p* = 0.085). Higher bilirubin in the HC compared to the LC subgroup remained throughout the 4-day DGA regimen until Day4 (Figure 4A). Conversely, Day0 blood albumin concentration was very similar in the HC and LC subgroups (Figure 4B). Interestingly, both blood albumin and bilirubin declined in the LC subgroup during the 4-day DGA regimen. Moreover, there existed a very tight association (*p* < 0.001) at an individual level between molar *changes* of albumin and bilirubin during the 4-day DGA regimen (Figure 4C). In the HC subgroup, there was no correlation between the changes of albumin and bilirubin during the 4-day DGA regimen (graph not shown). Furthermore, there was no association between the changes of Fe and albumin (R^2^ = 0.017, *p* >> 0.05) in the LC subgroup. We additionally measured the association between the 4-day changes of blood free fatty acids (= total fatty acids minus TGs) and blood albumin, and found only a mild positive correlation (R^2^ = 0.075, *p* > 0.05, graph not shown).

Remarkably, plasma alkaline phosphatase (ALP) was reduced in each LC participant during the 4-day DGA regimen (*p* < 0.0001, Figure 4D). Also, blood Fe concentration was reduced statistically very significantly in the LC subgroup (Figure 4E). The reduction of average blood Fe was 16.3% in the LC subgroup. Finally, there was a very strong correlation between the 4-day relative changes of blood Fe and bilirubin (Figure 4F, *p* < 0.0001).

### 3.5. 4-Day Responses of Blood Bilirubin and Fe Concentrations Are “Inversely” Associated with the Changes of Blood TGs and Subclinical Inflammation (GlycA)

Blood TGs concentration deviated strongly between the HC and LC subgroups at Day0 (Figure 5A). During the 4-day DGA regimen, the difference in TGs between the HC and LC subgroups reduced. Especially in the LC subgroup, there was a significant reduction in blood TGs. The Day0 concentrations in the HC and LC subgroups were opposite compared to each other in blood bilirubin (Figure 4A) and TGs. Nevertheless, the responses in bilirubin and TGs to the 4-day DGA regimen were in the same directions (Figure 4A and Figure 5A). Moreover, there existed a statistically significant positive correlation between the 4-day individual percent changes of blood TGs and blood bilirubin (Figure 5B), and with the 4-day percent changes of blood Fe (Figure 5C).

Blood subclinical inflammation (GlycA concentration) deviated strongly between the HC and LC subgroups at Day0 (Figure 5D). During the 4-day DGA regimen, the difference reduced and, especially in the LC subgroup, there occurred a remarkable reduction in blood GlycA (Figure 5D). There existed a statistically significant positive correlation at an individual level between the 4-day relative changes of GlycA and bilirubin (Figure 5E). Moreover, the 4-day relative changes of GlycA correlated strongly with the relative changes of Fe (Figure 5F). Finally, responses in GlycA during the 4-day DGA regimen resemble the respective changes in TGs (Figure 5A,D). There existed a very strong association between the 4-day relative changes of blood TGs and GlycA (R^2^ = 0.686, *p* < 0.001, graph not shown).

### 3.6. In Vitro Reduction of ROS

The DGA regimen effectively hinders excessive ROS generation and possesses a similar or superior equimolar antioxidant effect to glutathione, and vitamins C and E in hepatocytes (Appendix A). Similar ROS scavenging results were obtained from rat primary optic astrocytes, both in naturally induced and hydrogen peroxide induced ROS models (Appendix A).

## 4. Discussion

Blood bilirubin concentration increased significantly 12 h after the last DGA on Day21 in the DGA group but not in the placebo group. However, the increase in blood Fe and the mRNA expression of the HO-1 gene in WBCs were not statistically significant. As expected, the increase in blood bilirubin during the 21-day DGA regimen was associated with reduced blood TGs and systemic inflammation. In the acute 45 min test, HO-1 enzyme activity seemed to be activated, based on responses in blood bilirubin and Fe levels. In the 4-day comparisons intra-individual responses of blood bilirubin and albumin correlated very strongly in the LC subgroup, indicating an activation of hepatic bilirubin intake. Furthermore, hepatic inflow of Fe was possibly activated in the LC subgroup during the 4-day DGA regimen. Finally, the changes in blood TGs and GlycA between the days 4 and 21 correlated inversely with bilirubin (and Fe).

### 4.1. Acute 45 Min Activation of the HO-1 Pathway

Based on the increasing acute 45-min responses in blood bilirubin and Fe, it can be interpreted that the acute DGA dose seems to upregulate whole-body enzymatic activity of the HO-1 pathway. The suggestion is based on the fact that the HO-1 pathway is the only physiologically relevant source of bilirubin and Fe in blood [3,5,33]. Furthermore, statistically significant positive correlation between the changes of blood bilirubin and Fe was observed at the intra-individual level, which underlines that the main source of the acute responses is the same.

Interestingly, the acute increase in blood bilirubin concentration (+2.3%, Figure 3A) was much smaller than the 9.2% increase in blood Fe. At the same time, multistep HO-1 pathway reactions produce one iron and one bilirubin per one heme substrate, and the molar concentrations of bilirubin (10.3 μM) and Fe (17.9 μM) in blood on Day4 morning were rather close to each other. These somewhat contradictory results may be explained by a temporary increase in intracellular biliverdin/bilirubin concentration during the 45 min after the acute dose of DGA.

The increase in tissue biliverdin/bilirubin concentrations may materialize in the peripheral tissues that generate biliverdin from the HO-1 reaction [1] and/or in the liver that handles the excretion of bilirubin from the body [3,4]. In the latter case, liver sinusoids import bilirubin from the blood rapidly, which may be possible because after the 4-day DGA regimen the hepatic import of bilirubin was increased. In both cases, a possible cause of the increased concentrations is the intracellular antioxidant properties of the BVR enzyme activity [5]. In a reverse reaction, BVR is able to convert formed bilirubin back to biliverdin and induce a reducing antioxidant reaction in each cycle [5]. The acute DGA dose increases DGA concentration in cells which induces similar pathways as in exercise [27], while exercise signaling may induce transient ROS generation [20]. Scavenging of formed extra ROS may need antioxidant BVR activity and thus postpone the release of bilirubin into blood circulation [23] after the acute DGA dose.

### 4.2. Day21 Non-Acute Activation of the HO-1 Pathway

An important take away from the acute 45 min experiment is the transient activation of the HO-1 pathway. It happens after each DGA dose, even after the priming with earlier DGA doses. This pattern may eventually lead to the permanent (non-acute) upregulation in blood bilirubin and at least temporary whole-body enzymatic activation of the HO-1 pathway after each DGA dose. The 34% increase, although statistically non-significant, in the mRNA expression of HO-1 in WBCs was in line with transient whole-body activation of the HO-1 pathway.

We cannot exclude the possibility that the strong increase in blood bilirubin level from Day0 to Day21 in the DGA group was partially based on a downregulation of the hepatic conjugation activity of bilirubin. It has been shown that exercising persons possess elevated bilirubin levels [21], and further that rats with higher exercise capacity possess lower hepatic conjugation activity and thus higher plasma bilirubin concentration [22]. The DGA regimen activates similar pathways as an exercise regimen, e.g., the concentration of blood beta-hydroxybutyrate increases by 20% [27]. Thus, the observed almost 20% increase in bilirubin concentration, and also the higher bilirubin in the HC subgroup (Figure 4A), were consistent with earlier findings [23]. Interestingly, PPARα is one of the key transcription factors taking part in hepatic beta-hydroxybutyrate biosynthesis [34] and as already mentioned, bilirubin has been shown to activate PPARα nuclear translocation [8].

### 4.3. Day21 Non-Acute Activation of Downstream Metabolites and Pathways

Transient HO-1 pathway activation and bilirubin independently possess antioxidant and anti-inflammatory effects [5,10]. There existed a statistically significant association (*p* = 0.015) between the relative changes of bilirubin and GlycA from Day0 to Day21; the higher the individual rise in blood bilirubin, the greater is the reduction in individual subclinical systemic inflammation. Furthermore, relative changes in IL-6 also correlated negatively with the changes of bilirubin from Day4 to Day21. These results indicate that the positive anti-inflammatory effects of the long term DGA regimen may be partially mediated by the HO-1 pathway. Based on the present findings, it seems that the blood bilirubin level plays a direct role in the reduction of systemic inflammation. Persistent elevated systemic inflammation is often the cause of many chronic diseases [1] and thus its reduction may be of utmost importance.

According to a recent study, bilirubin activates cellular fatty acid use via the PPARα pathway [8]. The increase of blood bilirubin tends to reduce excretion of bilirubin from tissues and thus intracellular concentrations rise. The increase in bilirubin concentration should increase cellular use of TGs for mitochondrial fatty acid oxidation. Indeed, there was a statistically significant correlation between the relative changes in bilirubin and TG from Day0 to Day21. The higher the individual rise in blood bilirubin, the greater is the reduction in individual blood TGs concentration. Increased use of fatty acids in energy metabolism can be important in NAFLD [35].

Finally, HIF-1α expression is induced by bilirubin in the physiological oxygen content, i.e., normoxia [15]. Also, HO-1 activation and related CO generation may stabilize HIF-1α and result in cytoprotection [11]. Current human studies were performed at rest showing that HIF-1α mRNA expression was strongly upregulated in WBCs on Day21. This was well in line with bilirubin upregulation. Furthermore, the increase in HIF-1α mRNA upregulation by 49% in WBCs from Day4 to Day21 was statistically very significant and again in line with the statistically very significant upregulation of bilirubin during the same period. However, we could not find a statistically significant correlation between the changes of blood bilirubin level and the changes of HIF-1α mRNA expression in WBCs in the DGA group. Furthermore, HIF-1α intracellular proteasomal degradation is strongly increased in normoxia situation [36]. Thus, the strongly increased mRNA expressions of HIF-1α could also compensate for the increased HIF-1α enzyme degradation rate due to the improved inflammatory state in the DGA group (Figure 2D). Nevertheless, HIF-1α mRNA upregulation on Day21 was consistent with energy metabolic activation by the DGA regimen [27], which also likely caused the upregulation of the HO-1 pathway [10]. It may even be that the energy metabolic activation by the DGA regimen induced a marginal additional need for oxygen that gradually induced HIF-1α upregulation [17,37].

### 4.4. Four-Day Reduction in Bilirubin in the LC Subgroup Is Directly Associated with the Change in Plasma Albumin

There existed a difference in blood bilirubin concentrations between the HC and the LC subgroups on Day0 and the difference remained throughout the 4-day DGA regimen. Average bilirubin concentration was clearly higher in the HC subgroup. As already said, higher bilirubin possesses antioxidant effects [5,21], thus the HC subgroup is in a better position in this respect. Why the blood bilirubin concentration was reduced by the 4-day DGA regimen in the LC subgroup is still without answers.

Albumin serves as a carrier of blood unconjugated bilirubin. It also facilitates hepatic intake of unconjugated bilirubin. During the process of hepatic intake, the albumin-bilirubin complex is bound to the sinusoidal surfaces of hepatocytes and the process is relatively time consuming. Nevertheless, the albumin-bilirubin complex is not in blood circulation. Thus, when there is a change in the host’s hepatic demand for bilirubin from blood, it is expected to be reflected as a simultaneous change in blood albumin. Furthermore, the liver functions as the final destiny of blood bilirubin to be subsequently conjugated and excreted from the body with bile acids [4].

Blood bilirubin level declined by 9.6% in the LC subgroup during the 4-day DGA regimen, and albumin declined only by 3.0%; however, while this change in albumin is not physiologically relevant, it was nevertheless statistically very significant. Furthermore, there existed a very strong association (R^2^ = 0.705) between the 4-day molar changes in bilirubin and albumin in the LC subgroup. Because one albumin protein transports several bilirubin molecules [3,38], it seems clear that the very small relative reduction in the abundant pool of blood albumin is directly related to the hepatic inflow of bilirubin. All in all, after only eight doses of DGA in four days, it seems that hepatic demand for blood unconjugated bilirubin increased significantly in the LC subgroup. This result would be in line with the findings of our earlier study on the main energy metabolites, in which we demonstrated that hepatic demand for glycerol (and alanine) was strongly activated during the 4-day DGA regimen [27]. Interestingly, glycerol metabolism is positively regulated by PPARα in the liver [39].

### 4.5. Also, Other Bilirubin Related Blood Biomarkers Indicate Strong Hepatic Activation in Four Days

Blood Fe declined strongly in the LC subgroup during the 4-day DGA regimen. Furthermore, there was a very strong association (*p* < 0.0001) between the relative changes in bilirubin and Fe in blood. These two events indicate that hepatic iron demand was also increased [38]. However, it should be noticed that also HO-1 pathway downregulation could explain, at least partly, the strong correlation between the changes in bilirubin and Fe [1]. The strong 16.3% reduction in Fe in the LC subgroup may be additionally explained by the mitochondrial activation during the 4-day DGA regimen [27]. Mitochondrial activation increases the use of Fe in OXPHOS heme proteins [33]. This is because the liver especially seems to be activated significantly during the 4-day DGA regimen [27], and the demand of blood Fe for mitochondrial heme protein synthesis may have increased the most in the liver [40]. The blood Fe pool is only 3–4 mg and must turn over several times daily to meet the demand, mostly from erythropoiesis [33]. All in all, hepatic activation could explain most of the strong decline in blood Fe during the 4-day DGA regimen.

The four-day changes in blood TGs were also associated with the changes in Fe (R^2^ = 0.446, Figure 5C). This surprisingly strong relationship may indicate that the hepatic activation [41] in the LC subgroup may also be behind the decline in blood TGs in the LC subgroup. Change in blood TGs concentration is directly related to the net release of TGs from the liver [42], and a more active liver consumes a bigger part of formed TGs itself [43].

Finally, unconjugated bilirubin presumably flows into the liver with albumin and at the same time also the HO pathway was likely activated in the liver. To avoid excessive concentration of potentially toxic bilirubin, it is likely that hepatic conjugation and excretion of the conjugated bilirubin increased simultaneously. Conjugated bilirubin is excreted from hepatocytes as bile pigment via the bile ducts. ALP may be increased even if only a few small bile ducts are obstructed, and serum bilirubin is normal [44,45]. Remarkably, plasma ALP was reduced in each LC participant during the 4-day DGA activation (*p* < 0.0001). The extremely consistent reduction in ALP and the reduction in blood bilirubin in the LC subgroup may point to some unspecified hepatobiliary enhancement, such as the bile acid secretion to the small intestine [3], during the 4-day DGA regimen in the LC subgroup.

### 4.6. In Four Days, Systemic Inflammation (GlycA) and TGs Correlated Inversely with Blood Bilirubin

The 21-day changes in blood bilirubin and GlycA correlated negatively. During the 4-day DGA regimen, correlation between the changes in bilirubin and GlycA was positive, i.e., totally opposite to that on Day21. A technical explanation for this inverse 4-day bilirubin–GlycA correlation arises from the strong reduction of systemic inflammation in the LC subgroup and simultaneous modest increase in the HC subgroup, and from similar subgroup dynamics in blood bilirubin.

What happens during the 4-day DGA regimen that temporarily distorts the canonical relationship that has been observed also in the literature [1,23,24]? We do not know the exact reason, but the hepatic activation that is accompanied with the reduction in systemic low-grade inflammation may be an important explanatory factor during the 4-day DGA regimen. Additionally, it is possible that in the LC subgroup, the HO-1 pathway activity was reduced due to reduction in inflammation, and as a follow-up, the release of bilirubin and Fe from tissues to blood was reduced causing the inverse correlation. All in all, the positive 4-day correlations between GlycA and bilirubin, and GlycA and Fe can be likely explained by the activation of the liver and the downregulation of whole-body HO-1 pathway activity due to reduction of systemic inflammation.

Also, the correlation between the changes in bilirubin and TGs during the 4-day DGA regimen was positive, i.e., fully the opposite than during the period from Day0 to Day21. Partial explanation for the positive correlation likely arises from the presumed inflow of bilirubin into the liver in the LC subgroup. It possibly increased hepatic fatty acid oxidation via PPARα [8] and tended to reduce blood TGs more in participants with higher hepatic bilirubin inflow from blood, and vice versa. Further explanation for the positive correlation between bilirubin and TGs during the 4-day DGA regimen arises from the strong correlation between the changes in TGs and GlycA during the same 4-day period (R^2^ = 0.686), i.e., the correlations between the changes in blood TGs, bilirubin, and Fe were partially due to very strong association between the changes in GlycA and TGs.

### 4.7. DGA Regimen Hinders Excessive ROS Generation under Metabolic Stress in Hepatocytes and Astrocytes

The DGA regimen effectively hindered excessive ROS generation (Appendix A) [46]. The exact reason why the two-day DGA priming and acute DGA dose with fresh cell culture media 2 h earlier is able reduce net ROS generation remains elusive, but it is likely that the activation on the HO-1 pathway plays an important role in both hepatocytes [9] and astrocytes [7].

## 5. Conclusions

The inducible heme oxygenase pathway was activated transiently after each DGA dose during the 21-day study period. Simultaneously, blood bilirubin level was strongly upregulated towards Day21 under the DGA regimen. The positive associations of blood bilirubin and systemic inflammation (GlycA and IL-6) and blood lipids (TGs) were observed. Also, HIF-1α mRNA expression was shown to be strongly activated in WBCs by the 3-week DGA regimen. All these activations likely relate to the mitochondrial energy metabolic activation by the DGA [27].

During the first four days under the DGA regimen, the reduction in blood TGs in the LC subgroup may be partially linked to the hepatic inflow of bilirubin with plasma albumin. Simultaneously, subclinical inflammation was markedly improved in the LC subgroup, i.e., in the participants with somewhat elevated systemic inflammation before the study. Fast improvement of the anti-inflammatory status likely reduced HO-1 pathway activity in four days in the LC subgroup.

It seems that the DGA regimen is able to both up- and downregulate the HO-1 pathway in humans. This possibly materializes via temporary activation of ROS generation by OXPHOS after each DGA dose, and a more permanent reduction of oxidative stress inter alia via HO-1 pathway activation. In vitro studies with hepatocytes and astrocytes showed that the DGA regimen efficiently reduces ROS generation in metabolic stress.

All in all, the DGA regimen seems to be able to activate whole-body aerobic energy metabolism without excessive ROS generation. Benefits materialize in the whole body, but the liver is activated the most due to its central role in maintaining glucose homeostasis and its other vital homeostatic tasks. Increased plasma bilirubin and beta-hydroxybutyrate, and the activation of the HIF-1α pathway are beneficial also for neurodegenerative diseases. The therapeutic combination activated by the DGA regimen poses benefits for chronic diseases related to elevated systemic inflammation.

## Figures and Tables

**Figure 1 antioxidants-11-02319-f001:**
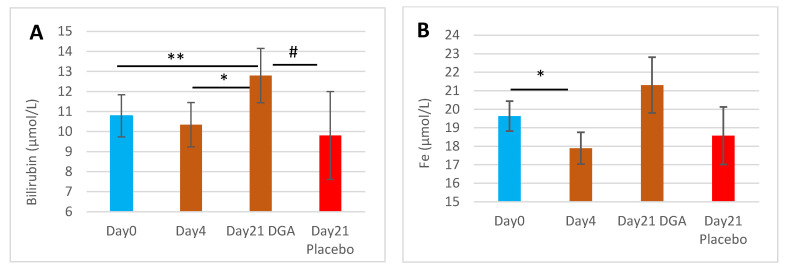
End products of the HO-1 pathway in blood 12 h after the last DGA or placebo dose: (**A**) Mean (±SEM) blood bilirubin concentration, (**B**) Mean (±SEM) blood Fe concentration. **Notes:** (1) The Day21 DGA and Placebo averages are indexed to Day0 so that Day21 fully reflects the change from Day0. (2) * and ** indicate paired *t*-test, # indicates non-paired *t*-test between the%-changes from Day0 to Day21.

**Figure 2 antioxidants-11-02319-f002:**
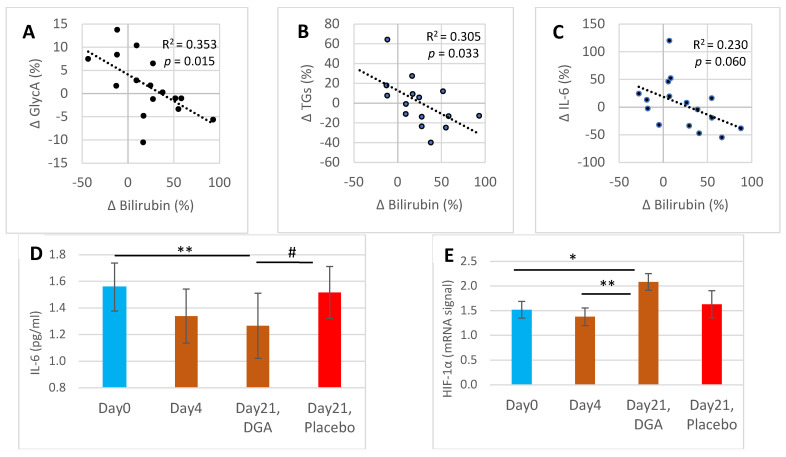
Bilirubin associated Day21 anti-inflammatory, fat metabolic, and mRNA expression effects: (**A**) scatter diagram and R^2^ of the percent changes from Day0 to Day21 between GlycA and bilirubin, (**B**) scatter diagram and R^2^ of the percent changes from Day0 to Day21 between blood triglycerides and bilirubin, (**C**) scatter diagram and R^2^ of the percent changes from Day4 to Day21 between IL-6 and bilirubin, (**D**) mean (±SEM) blood IL-6 concentration, (**E**) mean (±SEM) mRNA expression of HIF-1α from collected WBCs. **Notes:** (1) In (**D**,**E**), the Day21 DGA and placebo averages are indexed to Day0 so that Day21 fully reflects the change from Day0. (2) * and ** indicate paired *t*-test, # indicates non-paired *t*-test between the %-changes from Day0 to Day21. In (**A**–**C**), the *p*-values are based on an F-test. (3) One Day21 observation was deducted from (**B**) as an outlier. Additionally, one Day0 IL-6 observation was excluded from (**D**) when comparing the changes from Day0 to Day21 in the placebo group against the DGA group.

**Figure 3 antioxidants-11-02319-f003:**
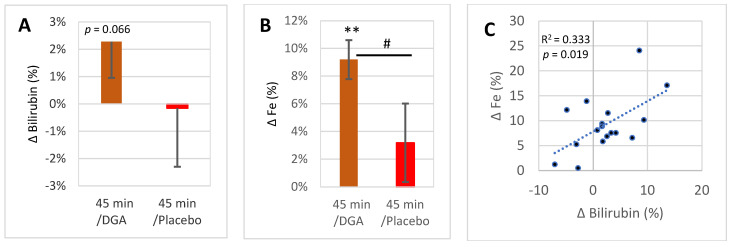
HO-1 pathway end products excluding CO in blood 45 min after acute DGA dose: (**A**) 45 min average intraindividual percent change in bilirubin (±SEM), (**B**) similar 45 min change in Fe (±SEM), (**C**) scatter diagram between 45 min %-changes of bilirubin and Fe after acute DGA dose. **Notes:** ** indicate paired *t*-test, # indicates non-paired *t*-test between the %-changes in 45 min. In (**C**), the *p*-value is based on an F-test.

**Figure 4 antioxidants-11-02319-f004:**
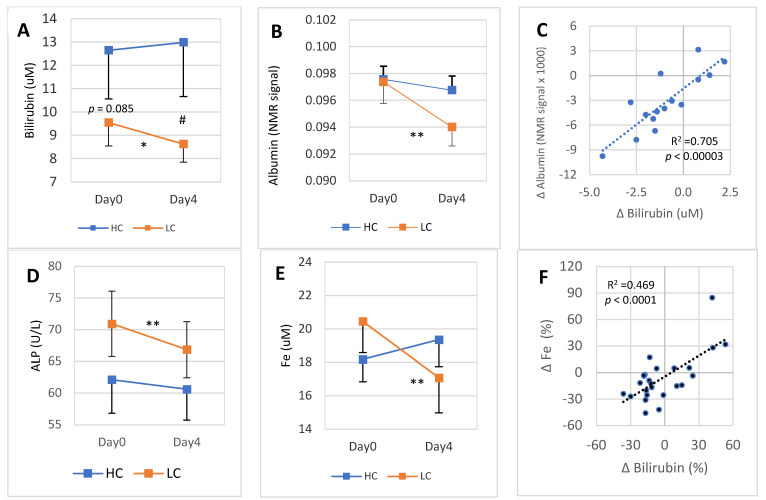
4-day DGA regimen activates hepatic bilirubin metabolism and Fe storage in the LC subgroup: (**A**) mean (±SEM) blood bilirubin in the HC and LC subgroups separately, (**B**) mean (±SEM) blood albumin in HC and LC subgroups separately, (**C**) cross correlation between intraindividual 4-day molar changes of albumin (vertical scale) and bilirubin (horizontal scale) in the LC subgroup. (**D**) mean (±SEM) blood ALP in the HC and LC subgroups separately, (**E**) mean (±SEM) blood Fe in the HC and LC subgroups separately, and (**D**) cross correlation between intra-individual four-day percent changes of Fe (vertical scale) and bilirubin (horizontal scale) in the whole group. **Notes:** * and ** indicate paired *t*-test from Day0 to Day4, # indicates non-paired *t*-test between the bilirubin concentration on Day4. In (**C**,**F**), the *p*-values are based on an F-test.

**Figure 5 antioxidants-11-02319-f005:**
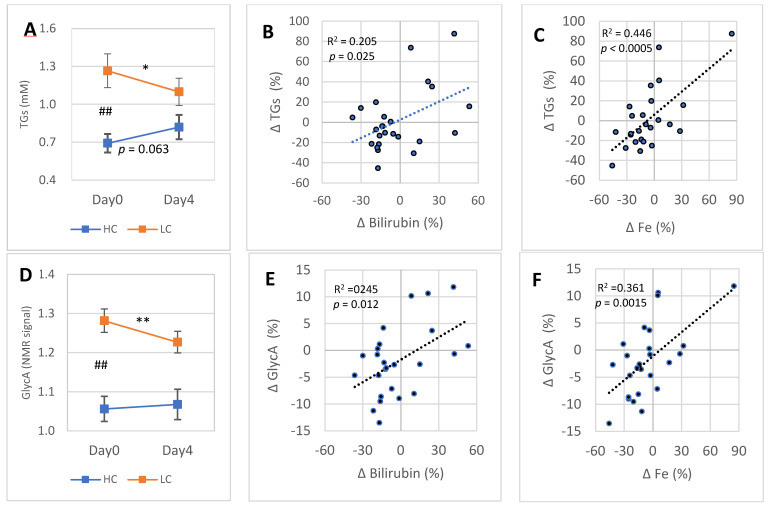
4-day DGA regimen activates the LC subgroup: (**A**) mean (±SEM) blood TGs in the HC and LC subgroups separately, (**B**,**C**) cross correlation between intra-individual four-day percent changes of TGs (vertical scale) and (**B**) blood bilirubin (horizontal scale) and (**C**) blood Fe (horizontal scale) in the whole group. (**D**) Mean (±SEM) blood GlycA in the HC and LC subgroups separately. (**E,F**) Cross correlation between intraindividual four-day percent changes of GlycA (vertical scale) and € blood bilirubin (horizontal scale) and (**F**) blood Fe (horizontal scale) in the whole group. **Notes:** * and ** indicate paired *t*-test from Day0 to Day4, # indicates non-paired *t*-test between the TGs and GlycA concentrations on Day0. In (**B**,**C**) and (**E**,**F**), the *p*-values are based on an F-test.

**Table 1 antioxidants-11-02319-t001:** (**A**) Study phases and timings, (**B**) characteristics of the study group and analysed subgroups modified from our earlier publication [27].

**(A)**					
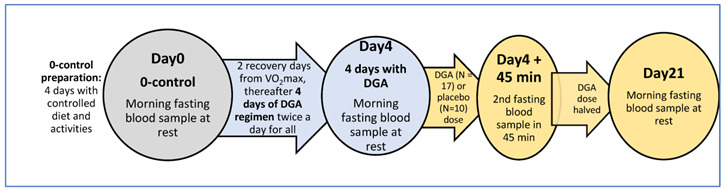					
**(B)**					
	**0-Control**	**4-Day Health, HC-LC**		**DGA vs. Placebo**	
**Study Group**	**Whole Group (*N* = 27)**	**LC Subgroup (*N* = 17)**	**HC Subgroup (*N* = 10)**	**DGA Group (*N* = 17)**	**Placebo Group (*N* = 10)**
Mean age (years)	56 years (range 50.3–60.9)	55	56	56	55
Mean VO_2_max (ml/kg/min)	35.5 (range 21.8–48.8)	31.3	43.8	35.1	36.1
Mean BMI	25.3 (range 20.1–31.7)	26.2	23.5	25.0	25.8
Female/Male	16 females/11 males	10/7	6/4	10/7	6/4

## Data Availability

The original data contributions presented in the study are included in the graphs of the article. Anonymous data on presented biomarkers is available from the corresponding author upon reasonable request. Additional anonymous data can be found in Hirvonen et al., (2021) [27] Supplement A.

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
