# Peer review of "Heme Oxygenase-1 and Blood Bilirubin Are Gradually Activated by Oral D-Glyceric Acid"

_antioxidants, 2022, doi:10.3390/antiox11122319_

Round 1
Reviewer 1 Report
Overall, the manuscript is well-written, and the study is well-performed. There are no major concerns. There are a few minor comments below that were noticeable when reviewing the manuscript.
· The second sentence of the abstract is written awkwardly. The ‘in 45 min after 4-day DGF regimen’ could be rewritten to have a better flow.
· The use of the wording ‘blood bilirubin’ throughout is okay in several places, but in some areas, it is awkwardly written, which breaks up the flow of the readership for the paper.
· Table 1 legend should be on the page with the table.
· The organization of some of the figures is spaced out too much, and some of the figures are overlapping. Please go through and make adjustments to improve the organization and spacing.
· The stats used in the figures should be mentioned in the figure legend.
· Some of the references in the discussion have a comma between them; please correct these.
· In several places, the authors write that DGA induces heme oxygenase activity, such as the statement in the conclusions “Inducible heme oxygenase pathway was activated transiently after each DGA dose 538 during the 21-day study period.” This is an overstatement as the measurements did not measure bilirubin production but measured levels. If the UGT1A1 enzyme were suppressed by DGA, then that would also increase blood bilirubin levels, and this might have happened without activating heme oxygenase. Please rework these sentences to reflect that DGR ‘regulates’ blood bilirubin levels rather than heme oxygenase activity. This will reflect the findings.
· The authors wrote, “Increased plasma bilirubin and beta-hydroxybutyrate, and the activation of HIF-1α pathway are beneficial also for the neurodegenerative diseases.” A previous study showed that bilirubin increases beta-hydroxybutyrate, which could be discussed in the text.
Reviewer 2 Report
In present study, authors were investigated the effects of oral DGA on heme-oxygenase-1 and blood bilirubin, and various other profiles. Particullary, these hypothesis were confirmed from middle ages of volunteers and well designated experimental methods and comparable parameters. So, authors provided valuable results including HO-1 pathway- and side effects-related parameters and suggested that DGA may reduce the risks of chronic diseases such as age-dependent disease like Parkinson’s diseases
In this article, authors were divided two separated experiments that administration/placebo and high aerobic capacity (HC)/lower aerobic capacity (LC) subgroups. As a results, maybe authors also deeply concerned, shown opposit both experimental data in some parameters. Therefore, authors should discuss more this with other possible hypothesis with current ‘healthy condition’ of volunteers.
Reviewer 3 Report
The overall design of the study is somewhat confusing. Line 352 the authors state "Blood bilirubin concentration increased significantly 12 h after the last DGA on Day21 in the DGA group but not in the placebo group." Couldn't the authors just state blood bilirubin concentration increased significantly on Day 21? I'm not sure where the 12 hours after the last DGA fits in.
The authors seem to suggest that elevations in HO-1 may at least explain the acute effects of DGA on serum bilirubin and iron. What site of HO-1 induction would be responsible for this? Hepatic? Splenic? other tissues?
The data in Figures 1 and 4 is somewhat confusing. Is the data in figure 1 the combined data for the HC and LC groups? This is not clear.
Do the authors know of any cardiovascular or metabolic effects of DGA treatment? Any effects on blood pressure or insulin resistance? Has this ever been tested?
